# Predictors of Quality of Life in HIV-Infected Persons from Mozambique: The Dual Role of Schooling

Jorge Lufiande [1], Susana Silva [1,*] , Ana Catarina Reis [2] and Marina Prista Guerra [1]

[1] Center for Psychology, Faculty of Psychology and Educational Sciences, University of Porto, Rua Alfredo Allen s/n, 4200-135 Porto, Portugal; jrglufiande33@gmail.com (J.L.); mguerra@fpce.up.pt (M.P.G.)

[2] São João University Hospital Center, Alameda Prof. Hernâni Monteiro, 4200-319 Porto, Portugal; anacatarinareis@outlook.com

* Correspondence: susanamsilva@fpce.up.pt

**Abstract:** Increasing quality of life (QoL) is both an end in itself and a means to optimize the impact of treatment in HIV-infected persons. Possibly due to cultural and social influences, the predictors of QoL vary across studies, highlighting the importance of studying specific populations. In the present study, we aimed to determine the sociodemographic (age, sex and schooling, or number of years at school) and psychosocial correlates (meaning in life, social support, positive and negative affects) of QoL in HIV-infected persons living in Mozambique, a country with a high prevalence of HIV but also with well-structured strategies to fight the disease. To that end, we made correlational analyses followed by regression models and examined potential mediation processes among predictors. All correlates were relevant except for sex. Meaning in life was the strongest predictor, while social support was the weakest. Schooling was both directly and indirectly related with QoL—in the latter case, it was mediated by meaning in life, social support and positive affect. Our findings suggest that investments in education may be highly rewarding to Mozambicans, and that satisfying needs for self-actualization and purpose may be more urgent than improving social connections.

**Keywords:** HIV/AIDS; Mozambique; quality of life; schooling; meaning in life; social support; affect balance



## 1. Introduction

Available records indicate that 38.4 million people (0.7% of the world's population) were infected with HIV (human immunodeficiency virus) in 2021 [1]. Sub-Saharan Africa encompasses two thirds of new infections, and Mozambique ranks highly on the list of burden HIV countries [2]. In 2021, the prevalence of HIV infections reached 12.5% (one in eight citizens were older than 15 years; 15% in women and 9.5% in men). The provinces most affected were Gaza (prevalence of 20.9%) and Zambezia (17.1%). Although many challenges remain unaddressed, there has been great progress regarding treatment, and therapeutic adherence in Mozambique is currently around 95% [3]. Antiretroviral drugs increased the life expectancy of HIV-infected persons and transformed the disease into a chronic condition that can be managed [4]. These drugs may reduce viral load to undetectable levels, promoting a sharp decrease in mortality rates. Antiretrovirals enhance the quality of life of HIV-infected persons [5,6], but they may also cause damage—to both their physical conditions (secondary effects of treatment) and to other domains of quality of life [7].

Quality of life (QoL) is a subjective and relative measure [8] that the World Health Organization describes as "individuals' perceptions of their position in life in relation to their goals, expectations, standards and concerns, in the context of the culture and value system in which they live" ([9], p. 16). QoL levels can be markedly weak in HIV-infected persons, even when compared to other chronic diseases [10], and they have been used to

monitor these individuals. QoL levels provide a comprehensive understanding of the damage caused by the disease and its treatment, thus allowing personalized multidisciplinary interventions [11].

The relation of QoL with sociodemographic and/or psychosocial variables has been investigated in several studies [12–15] as a way to identify potential influences on the QoL of HIV-infected persons. Within sociodemographic correlates, some studies indicate that QoL may increase with schooling—the total number of years spent in school to obtain an education—and decrease with age [12,16]. Findings related to age are nevertheless mixed: null results [15] as well as positive associations (increased QoL in older patients, e.g., [17]) have also been reported. Findings related to sex are also not consensual. While some studies reported increased QoL in women [14], others observed an advantage in men [12,16], with the latter highlighting cultural influences. Reports of sex-related differences restricted to certain domains [15,16] are also available.

Along with sociodemographic variables, psychosocial correlates, such as social support, meaning in life or affect balance, have also been reported as potentially relevant influences on the QoL of patients in general [17] as well as regarding HIV infection specifically [18]. Social support describes the extent to which social connections help mitigate the negative consequences of disease [19]. Social support is essential to stress management in health-related crises [20] and, specifically, to the dimensions of HIV management [21–23] such as prevention, adherence to treatment and recovery [20]. Several studies have highlighted the association between social support and QoL in disease (e.g., [24]). As for meaning in life, it quantifies the strength of an individual's sense of purpose and their drive towards goal attainment, and it may include a dimension of altruism [25]. Meaning in life is related to QoL in healthy (e.g., [26]) as well as in HIV-infected individuals [15,18,27,28]. Finally, available instruments to evaluate affect balance measure how often participants experience negative (e.g., guilt, shame) vs. positive affect (e.g., strength, determination) affect [29]. Research shows that affect balance is associated with QoL in various contexts [30].

Although some findings on the correlates of QoL are replicated across studies, the fact that these are permeable to cultural influences (e.g., in countries with low population density, social support may be more important) highlights the need to investigate this topic in specific contexts. In addition, the majority of studies do not take into account possible mediation processes that may lead to misleading conclusions. Mediation occurs when two different predictors seem to be independent but, when gathered in a single model, one of these loses relevance because the other explains its association with the dependent variable [31]. In the present study, we examined the correlates of QoL in HIV-infected persons living in Mozambique—a poor country which has very high prevalence of HIV but is also highly committed to tackling the problem. To that end, we considered a set of potentially relevant sociodemographic and psychosocial variables and analyzed their associations with QoL. Based on this, we built regression models and verified whether the associations predictor-dependent variable remained significant. When this did not happen, we examined whether mediation was present.

## 2. Materials and Methods

### 2.1. Participants

A total of 352 HIV-infected persons living in Zambezia (one of the most affected provinces in Mozambique) agreed to take part in this study. To be included, participants should (1) be receiving retroviral treatment for at least six months, (2) have gone to school, (3) be older than 17 years and (4) be free of severe mental health pathologies.

As shown in Table 1, the age range of participants was wide (18–59 years) and the majority were women (62%). Mean schooling (9 years) was clearly below university level. In addition, most participants were unemployed (71.5% unemployed; 25.4% employed; and 3.1% students) and lived with a partner (71.3%). The vast majority came from two specific districts (Quelimane and Mocuba) from the province of Zambezia (94%). Some (11.7%) had no permanent address.

**Table 1.** Sociodemographic characteristics of participants.

|  | Minimum | Maximum | Mean | Standard Deviation |
|---|---|---|---|---|
| Age (years) | 18 | 59 | 35.17 | 9.851 |
| Schooling (years) | 1 | 25 | 9.93 | 2.76 |
| **Sex** |  |  | **N** | **%** |
|  |  | Men | 135 | 38.4 |
|  |  | Women | 217 | 61.6 |

Regarding clinical variables, the sample was relatively homogeneous (see Supplementary Table S1, showing the clinical characteristics of participants). All patients were carriers of HIV 1, and most of them (88.3%) were unaware of the source of contagion. In the vast majority, the infection was controlled (86.4% with <50 copies of the virus—highest count would be >30,000). Most participants (96.3%) had never interrupted treatment, and 88.4% had remained on the same medication since they began therapy. Nearly 85% of participants reported feeling no secondary effects from antiretroviral drugs.

### 2.2. Instruments

To characterize QoL, we used the WHOQOL-Bref (World Health Organization Quality Of Life—Bref), a short version of the self-report questionnaire WHOQOL-100 [32], validated for the Portuguese population by [33]. The instrument comprises 26 items, organized into four domains: physical, psychological, social and environmental. Responses are provided on Likert scales, referring to intensity, capacity, frequency and evaluation. In the current study, the Cronbach's alpha for all items was 0.90. Reliability values for physical and psychological domains were higher (0.80) than for social (0.57) and environmental (0.67) domains.

Psychosocial variables were self-reported using three different instruments: the Social Support Scale (Escala de Suporte Social, ESS [34]), the Meaning in Life Scale [18] and the Positive and Negative Affect Schedule [35], adapted to Portuguese by [29]. The Social Support Scale comprises 20 items organized into five dimensions: socio-affective, financial, familiar and romantic support and freedom from external control. Participants responded on a 5-point Likert scale, ranging from "quite dissatisfied" to "quite satisfied". The Cronbach's alpha in our sample was 0.73. The Meaning in Life scale is made up of seven items, some of these presented in an inverted form to avoid social desirability effects [25]. Participants respond on a 5-point scale, with scores ranging from 7 to 35. Reliability, as measured by Cronbach's alpha, was 0.58 in our study—lower than the value obtained by Reis et al. [18] in a sample of Portuguese HIV-infected persons. Finally, PANAS is composed of 20 items, 10 expressing positive affects—the propensity to experience positive emotions and interact with others positively (e.g., strong, determined)—and 10 expressing negative affects—the tendency to experience the world in a more negative way (e.g., guilty, afraid). Cronbach's alpha was 0.78 for the positive affect dimensions and 0.73 for negative affect dimensions.

### 2.3. Procedure

First, we requested ethical clearance for this project from the Ethical committee of University of Porto (FPCE-UP), which was given in May 2020 (Ref. 2020/04-1b). The decision was later ratified by the Bioethics Committee of Mozambique (Ref. 114/CIBS-Z/21, 13 August 2021). A request for data collection was then submitted to the district services of health, women and social action and consent was obtained.

We collected data at six local health units from Quelimane and Mocuba between September and December 2021. We approached participants while they waited for their appointments or prescriptions. Those who agreed to take part in the study provided informed consent according to the Declaration of Helsinki.

The questionnaires were administered inside the premises of health units, mainly with help from health technicians, as requested by participants, and with the main researcher

present in the room. The COVID-19 sanitary protocols were maintained. Clinical data were afterwards collected from the patients' files and electronic databases.

*2.4. Data Analysis*

We started with descriptive statistics for QoL and the three psychosocial variables in order to examine differences across QoL domains and compare the obtained values with those from other samples. To achieve our main goal, we began by analyzing the associations of QoL with sociodemographic and psychosocial variables using Pearson's correlations for continuous correlates and independent sample *t*-tests for sex. Based on these values, we defined regression models for each QoL domain (four models) using the enter method. The number of predictors in each model was dictated by sample size as well as by the magnitude of the associations seen before [36]. Please note that we use the word "predictors" when referring to regression results and "correlates" when approaching correlations. In both cases, we are not assuming a causal relation between variables, though we admit it may be present.

Results from regression models suggested the presence of mediation effects regarding the sociodemographic variable "schooling" (associations with QoL that were first observed vanished when other predictors were added). Therefore, we tested whether schooling effects were mediated by other variables.

We set the threshold for significance as $p < 0.05$. Analyses were performed with SPSS (IBM, New York, NY, USA) and JASP (University of Amsterdam, Amsterdam, The Netherlands), the latter used in mediation analyses. Assumptions for each test were previously checked.

## 3. Results

Section 3.1. provides descriptives for QoL and for the four psychosocial correlates (please see Section 2.1. for descriptives on sociodemographic correlates). Our main research question is addressed in Section 3.2. (correlations of QoL with sociodemographic and psychosocial variables), Section 3.3 (regression models based on the previous correlations) and Section 3.4. (mediation processes made visible when comparing correlations with regression results).

*3.1. Descriptives for Psychosocial Correlates and QoL*

Values for psychosocial correlates (Table 2) were similar to those obtained in Portuguese HIV-infected persons [13,18,37], including the increased weight of positive compared to negative affect. As for QoL, values were also generally similar to, or even higher than, the ones obtained in Portuguese [18,37] or Nigerian HIV-infected persons [16], with a single exception for the environmental domain, showing lower values than those found in one study with Portuguese participants [37].

**Table 2.** Descriptives for psychosocial correlates and quality of life (Qol) domains (possible range of scores in parenthesis).

| | Minimum | Maximum | Median | Mean | Standard Deviation |
|---|---|---|---|---|---|
| Psychosocial correlates | | | | | |
| Social support (20–100) | 46 | 89 | 70 | 69.06 | 9.22 |
| Meaning in life (7–35) | 17 | 34 | 25.00 | 25.17 | 3.50 |
| Positive affect (10–50) | 12 | 49 | 22.00 | 24.25 | 6.80 |
| Negative affect (10–50) | 10 | 35 | 20.00 | 19.96 | 5.49 |
| QoL domains | | | | | |
| Physical (0–100) | 35.71 | 100 | 64.28 | 67.86 | 15.45 |
| Psychological (0–100) | 16.67 | 100 | 70.83 | 67.88 | 17.48 |
| Social (0–100) | 16.67 | 100 | 66.66 | 64.48 | 14.84 |
| Environmental (0–100) | 25 | 90.63 | 56.25 | 58.44 | 11.46 |

### 3.2. Associations between Qol and Potential Correlates

As shown in Table 3, schooling correlated positively with all four QoL domains. Age showed negative correlations with the physical and psychological domains, and sex was not significantly associated with any QoL domain (*ps* > 0.11). All significant correlations were weak (<0.30).

**Table 3.** Correlations between quality of life (Qol) domains and sociodemographic variables.

| QoL Domains | Schooling | Age | Sex |
|---|---|---|---|
| Physical | 0.248 ** | −0.166 ** | n.s. |
| Psychological | 0.189 ** | −0.249 ** | n.s. |
| Social | 0.266 ** | n.s. | n.s. |
| Environmental | 0.270 ** | n.s. | n.s. |

** *p* < 0.01, n.s. non-significant.

As we expected (Table 4), negative affect correlated negatively with QoL, while the other three variables showed positive correlations. All correlations were significant, except the one between psychological QoL and social support. The associations of QoL with meaning in life and with positive affect were moderate (>0.30) to strong (>0.50), weak to moderate for negative affect, and weak for social support.

**Table 4.** Correlations between quality of life (Qol) domains and psychosocial variables.

| QoL Domains | Meaning in Life | Social Support | Positive Affect | Negative Affect |
|---|---|---|---|---|
| Physical | 0.661 ** | 0.186 *** | 0.515 ** | −0.307 ** |
| Psychological | 0.651 ** | n.s | 0.397 ** | −0.364 ** |
| Social | 0.459 ** | 0.326 *** | 0.463 ** | −0.109 * |
| Environmental | 0.550 ** | 0.232 *** | 0.418 ** | −0.270 ** |

* *p* < 0.05, ** *p* < 0.01, *** *p* < 0.001, n.s. non-significant.

### 3.3. Predictors of QoL per Domain

As presented in Table 5, the regression models for physical and psychological QoL explained more variance ($R^2_{Aj}$) than those created for social and environmental QoL, but all four models were significant. The strongest predictor was meaning in life, followed by negative and positive affect; however, it should be noted that negative affect did not predict social QoL values.

**Table 5.** Regression models for quality of life (Qol) domains ($R^2_{Aj}$ = $R^2_{adjusted}$; df = degrees of freedom; Part r = partial correlation).

| Model | β | t | p | $R^2_{Aj}$ | df | F | p | Part r |
|---|---|---|---|---|---|---|---|---|
| Physical QoL | | | | 0.531 | 6 | 65.56 | <0.001 | |
| Age | −0.129 | −3.404 | <0.001 | | | | | −0.125 |
| Schooling | 0.044 | 1.134 | 0.258 | | | | | 0.042 |
| Meaning in life | 0.430 | 8.715 | <0.001 | | | | | 0.320 |
| Social support | 0.095 | 2.382 | 0.018 | | | | | 0.087 |
| Positive affect | 0.252 | 4.980 | <0.001 | | | | | 0.183 |
| Negative affect | −0.242 | −6.146 | <0.001 | | | | | −0.226 |
| Psychological QoL | | | | 0.515 | 5 | 74.11 | <0.001 | |
| Age | −0.182 | −4.821 | <0.001 | | | | | −0.180 |
| Schooling | 0.005 | 0.121 | 0.904 | | | | | 0.005 |
| Meaning in life | 0.492 | 9.850 | <0.001 | | | | | 0.368 |
| Positive affect | 0.141 | 2.856 | 0.005 | | | | | 0.107 |
| Negative affect | −0.273 | −6.816 | <0.001 | | | | | −0.255 |

**Table 5.** *Cont.*

| Model | β | t | p | $R^2_{Aj}$ | df | F | p | Part r |
|---|---|---|---|---|---|---|---|---|
| Social QoL | | | | 0.324 | 5 | 34.43 | <0.001 | |
| Schooling | 0.112 | 2.419 | 0.016 | | | | | 0.107 |
| Meaning in life | 0.266 | 4.518 | <0.001 | | | | | 0.199 |
| Social support | 0.220 | 4.693 | <0.001 | | | | | 0.207 |
| Positive affect | 0.221 | 3.653 | <0.001 | | | | | 0.161 |
| Negative affect | −0.076 | −1.614 | 0.107 | | | | | −0.071 |
| Environmental QoL | | | | 0.380 | 5 | 43.58 | <0.001 | |
| Schooling | 0.101 | 2.293 | 0.022 | | | | | 0.097 |
| Meaning in life | 0.383 | 6.858 | <0.001 | | | | | 0.290 |
| Social support | 0.142 | 3.173 | 0.002 | | | | | 0.134 |
| Positive affect | 0.145 | 2.503 | 0.013 | | | | | 0.106 |
| Negative affect | −0.203 | −4.484 | <0.001 | | | | | −0.189 |

Schooling—which was significantly correlated with all QoL domains (see Table 3)—predicted social and environmental QoL, but associations with the physical and psychological domains vanished when schooling was inserted in the models together with other predictors. This pointed to the possibility of mediation effects [30], wherein any of these other predictors in the physical and psychological QoL models could be mediating the apparent effect of schooling.

*3.4. Mediation Effects*

In order to determine the potential mediators, we tested the correlations between schooling (independent variable) and the other correlates (potential mediators)—since significant values would be required for mediation to be considered. The association with age was non-significant ($p > 0.67$) and so was the association with negative affect ($p > 0.05$). In contrast, meaning in life ($r = 0.242$, $p < 0.001$), positive affect ($r = 0.234$, $p < 0.001$) and social support ($r = 0.138$, $p = 0.010$) showed significant positive correlations with schooling. Therefore, we examined the potential mediating role of meaning in life, social support and positive affect in the schooling–physical QoL association, and the role of meaning in life and positive affect in the association with psychological QoL (Table 6).

**Table 6.** Estimates for direct vs. indirect relationships of schooling with physical and psychological quality of life (Qol) as mediated by meaning in life, social support and positive affect.

| **Schooling and Physical QoL** | | | | |
|---|---|---|---|---|
| **Mediators** | **Direct** | **Indirect** | **Total** | **Mediation** |
| Meaning in life | 0.390 * | 0.640 *** | 1.020 *** | partial |
| Social support | 0.933 *** | 0.088 * | 1.021 *** | partial |
| Positive affect | 0.035 ** | 0.030 *** | 0.067 *** | partial |
| **Schooling and Psychological QoL** | | | | |
| **Mediators** | **Direct** | **Indirect** | **Total** | **Mediation** |
| Meaning in life | 0.160 | 0.720 *** | 0.880 *** | full |
| Positive affect | 0.026 * | 0.024 *** | 0.050 *** | partial |

\* $p < 0.05$, \*\* $p < 0.01$, \*\*\* $p < 0.001$.

For the physical domain, all three mediators showed significant values for both indirect (mediated) and direct effects, indicating partial mediation through meaning in life, positive affect (both $p < 0.001$) and, with less significance, social support ($p = 0.049$). For the psychological domain, positive affect partly mediated the schooling–QoL association (significant indirect and direct effects), while meaning in life conveyed full mediation (null direct effects with significant indirect effects).

In sum, age, social support, meaning in life, positive affect and negative affect were significant predictors of QoL in at least three of the four domains. Schooling seemed to be significantly related to QoL, but mediation analyses demonstrated that this association was partly (or, in one case, fully) mediated by psychosocial correlates.

## 4. Discussion

In the present study, our goal was to determine the sociodemographic (age, sex and schooling) and psychosocial correlates (meaning in life, social support, positive and negative affect) of QoL in Mozambican HIV-infected persons. To that end, we performed correlational analyses followed by regression models, and we examined potential mediation processes among predictors. We found that all correlates were relevant except sex, that meaning in life was the strongest predictor and that schooling was both directly and indirectly related to QoL—in the latter case, it was mediated by meaning in life and positive affect.

Regarding sociodemographic correlates, age correlated negatively with physical and psychological QoL, and it remained a significant predictor in the regression models for these two domains. This finding is in line with some studies on HIV [12,16] but contrasts with the null results [15] and the positive associations [17] that have also been found. One explanation for the negative correlations we found may be the loss of independence that comes with age, which would be particularly important to the psychological domain of QoL. Decreased physical QoL in older age groups is likely to be associated with biological aging. Also, in contrast to some studies (e.g., [16]), sex was not associated with QoL.

Contrary to some studies [15,17,37], associations between schooling and QoL were observed for all QoL domains. Further analyses showed that these associations were direct (i.e., non-mediated) for social and environmental domains, but indirect associations were also present in physical and psychological domains. For these, meaning in life, social support and positive affect mediated the association between schooling and QoL. In the psychological domain, indirect effects were striking, in that meaning in life fully explained the association. Direct effects—which were seen for physical, social and environmental QoL—may be accounted for by healthier habits, increased social outreach and more favorable economic perspectives of those with higher schooling levels. As for the indirect relations between schooling and QoL, these may relate to the increased sense of purpose and the joy of learning that may accompany the privilege of having an education in a country where school access is not generalized.

Concerning psychosocial correlates, positive and negative affects correlated positively and negatively with all QoL domains as we expected, with the exception of negative affect and social QoL, which showed no correlation. A similar scenario was seen in regression models. The significant associations we saw are in line with the literature [18,37], even though the domains where these associations exist vary across studies. Affect balance—the relationship between positive and negative affects—favored positive affects, which is also in line with the literature.

Regarding social support, results were somehow surprising, in that associations with psychological QoL were null, and the significant relations with the other three domains were weak, both in correlational and regression-based analyses. Social support not only optimizes perceptions and expectations regarding treatment as it also tends to have a positive emotional impact [38]. So, why was there such a weak link? One possibility is that basic social needs are already satisfied in Mozambique, due either to cultural traditions of community living or to social networks generated by current HIV policies. For instance, health units have implemented a "family file" system [39], where each patient has access to the information and prescriptions of any other family member also living with HIV. The high adherence to treatment may also act to strengthen a community of HIV-infected persons, where the sharing of experiences is facilitated. Therefore, in the face of a highly structured social network, patients may not be too sensitive to their particular social circumstances.

Finally, meaning in life showed moderate to strong positive correlations with all QoL domains, and the associations prevailed in the context of regression models. These findings replicate those from studies with healthy populations [26,40] and HIV-infected persons [16–18,37], even though [18] only saw significant associations with psychological and environmental QoL. These findings are in line with the idea that health crises tend to challenge the sense of purpose of an individual [41] and those who are equipped with the best tools to achieve psychological adjustment are those who overcome the challenge more easily.

Our study has limitations, and we would highlight four of these. First, even though the instruments we used were validated for the Portuguese population, there were none for the Mozambican people. Mozambicans speak Portuguese, but there are certain differences in the dialects, just as there are differences between European and Brazilian Portuguese. More critically, the instruments were not designed to take into account the specifics of the Mozambican culture. This may explain why we had low reliability values for certain measurements and suggests that it may be useful to promote proper validation in future research. Second, we did not use a control group of healthy citizens from Mozambique in this study. This prevents us from drawing solid inferences from our results, since comparisons between Mozambique and other countries may be mistaking HIV-infected persons' specificities with cultural specificities. Adding a control group should thus be a priority in future research. Third, the fact that we observed relatively high values for QoL in these HIV-infected persons—all undergoing treatment and with very low viral charge—is consistent with the literature [23,42], but it still raises one question: to what extent would these values replicate in patients with increased viral loads and little or no therapeutic adherence? In other words, it is likely that, beyond sociodemographic and psychosocial variables, the quality of health care plays an important role, and, in this sense, this variable is likely a strong predictor of QoL in HIV. Future studies could, therefore, add correlates such as viral charge and adherence to treatment to the potential predictors of QoL in HIV. Finally, we should note that we relied on the general approach to QoL, instead of assuming the more specific framework of health-related quality of life [43], which includes HIV-dedicated instruments to evaluate QoL [44]. We utilized this method because of constraints related to the larger-scale project in which this study is included and also because the latter approach is not consensual in the literature [45]. We believe, though, that the health-related QoL approach may provide complementary views on this topic and thus deserves future research attention.

## 5. Conclusions

Our study contributed to a better understanding of QoL in Mozambican HIV-infected persons in at least two ways. First, schooling seems to have a dual role: it enhances both QoL and psychosocial aspects like meaning in life, social support and positive affect—which, in turn, also increase QoL. From the viewpoint of practical applications, this suggests that investments in education may be highly rewarding. This would be valid for not only HIV-infected persons, who might see a greater impact of treatment or even increased mobilization for diagnosis, but also for all Mozambicans, who long to enhance their sense of purpose, social support and positive affect. Second, the weak contribution of social support to QoL, compared to the importance of education and meaning in life, suggests that self-actualization and purpose may be areas in Mozambican people's lives that need more attention than social connection.

**Supplementary Materials:** The following supporting information can be downloaded at: https://www.mdpi.com/article/10.3390/idr15040040/s1, Table S1: Clinical characteristics of participants.

**Author Contributions:** Conceptualization, J.L., A.C.R. and M.P.G.; methodology, J.L., A.C.R. and M.P.G.; formal analysis, J.L., S.S., A.C.R. and M.P.G.; investigation, J.L., A.C.R. and M.P.G.; data curation, J.L., S.S., A.C.R. and M.P.G.; writing—original draft preparation, J.L. and S.S.; writing—review and editing, J.L., S.S., A.C.R. and M.P.G.; visualization, J.L., S.S., A.C.R. and M.P.G.; supervision,

A.C.R. and M.P.G.; project administration, M.P.G.; funding acquisition, J.L. and M.P.G. All authors have read and agreed to the published version of the manuscript.

**Funding:** This research was funded by the Portuguese Foundation for Science and Technology, grant number UIDB/00050/2020 and by the Government of Mozambique via University of Licungo-Quelimane, grant (143.402). The APC was funded by the Portuguese Foundation for Science and Technology, grant number UIDB/00050/2020, and by the Government of Mozambique via the University of Licungo-Quelimane, grant (143.402).

**Institutional Review Board Statement:** The study was conducted in accordance with the Declaration of Helsinki and approved by the Ethics Committee of FPCEUP (2020/04-1b, 12 May 2020).

**Informed Consent Statement:** Informed consent was obtained from all subjects involved in the study.

**Data Availability Statement:** The database used in this study is available at the OSF link https://osf.io/32mtg/?view_only=ccb85d4e976d4edaa2dadb0a3779741f (accessed on 19 May 2023).

**Conflicts of Interest:** The authors declare no conflict of interest.

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
