# Peer review of "Predictors of Quality of Life in HIV-Infected Persons from Mozambique: The Dual Role of Schooling"

_2036-7449, doi:10.3390/idr15040040_

Round 1

Reviewer 1 Report

HIV is a global problem, but the paper lacks an international and African context.

Paper belongs to a separate group of quality of life research, which is Health Related Quality of Life. There is no mention in the text that such a thing exists. I recommend incorporating knowledge from Health Related Quality of Life. Headword. https://doi.org/10.1007/978-1-4419-1005-9_753.

The paper is focused on the quality of life of people with HIV. Does it differ from the quality of life of the rest of the population? I recommend incorporating knowledge from other authors, e.g. DOI: 10.1097/QAD.0000000000001672

The authors are from Portugal. How does the quality of life of people with HIV in Portugal differ from the quality of life of people with HIV in Mozambique? DOI: https://doi.org/10.3389/fpubh.2019.00266

Does the quality of life of people with HIV in Mozambique differ from the quality of life of people with HIV in other African countries? 10.1016/S2214-109X(17)30367-4.

The shortcoming of the paper is that the terms used, except quality of life, are not defined or at least explained. According to the title, the paper is focused on predictors, but the text uses the term "correlates". Is correlate the same as predictor? I recommend reading the paper

DOI: 10.3390/ijerph19106185.

The shortcoming of the paper is that it does not indicate the values of the quality of life of people according to gender, age and education.

The paper lacks a conclusion

Reviewer 2 Report

Lufiande et al. studied the quality of life of HIV infected population in Mozambique undergoing treatment. The test out different sociodemographic as well as psychosocial variables/factors to understand what has significant effect on the QoL. However, for any reader who hasn’t read the Portuguese article (Vaz-Serra et al, 2006), the authors should briefly describe the variables used in this study. Explaining them may avoid confusion, for example- what does “schooling” refer to? Does it mean education or literacy? Till what level/degree? Can the authors detail what are included in positive and negative effects?

I agree with the author that because of the social and cultural differences across countries or even areas within countries, the assessment and prediction of quality of life may vastly vary between populations. Hence, applying the predictors of QoL optimized for Portuguese population may be a limiting factor for Mozambique population, but this is a limitation of the field and authors can not correct that within the scope of this article. Nonetheless, it would be very interesting to find out difference in results when the same population is assessed by 2-3 predictors optimized for different populations.

Here are some minor comments that needs to be addressed in this article:

Line 38 and 40: References are added differently here compared to the rest of the article (words vs numbers).

Line 63: Does the author mean “her/his”?

Line 70-73: Please reword this sentence. Maybe dividing it into 2 sentences will help.

Line 88-92: Refer to table 1 in this paragraph.

Line 93-98: Kindly provide a supplementary table with the exact number of participants leading to these percentages.

Line 101: Kindly expand WHOQOL as it is used for first time here.

Line 112-118: Throughout the text, writing “0.XX” will be preferred in place of “.XX” to avoid confusion with “XX”.

Line 150: It will be beneficial to organize the result sections better- starting with the hypothesis/question, describing the data stepwise, and then ending with a one sentence summary of the inference from the results.

Line 159: Expand the abbreviations included in 1st row, in the table caption.

Line 164: How were the correlations determined to be significant of the were p>0.05?

Line 166 (under Table 3): Kindly add “0” before the decimals.

Line 171: Describe the range or what is considered to be “strong”, “medium”, and “weak”.

Article needs to be checked by a professional for grammatical errors and needs multiple corrections. For example, punctuation marks like comma and full stop are missing in multiple places. Some sentences are complicated and need rewording. At some places words are used wrongly, for example affect vs effect.

Round 2

Reviewer 1 Report

I have no comments

Author Response

Dear Reviewer,

Thank you very much for your time and effort while making a critical reading of our paper.